# Host-dependent resistance of Group A *Streptococcus* to sulfamethoxazole mediated by a horizontally-acquired reduced folate transporter

M. Kalindu D. Rodrigo [1], Aarti Saiganesh [1], Andrew J. Hayes [2], Alisha M. Wilson [1], Jack Anstey [1], Janessa L. Pickering [1], Jua Iwasaki[1], Jessica Hillas[1], Scott Winslow[1], Tabitha Woodman [1], Philipp Nitschke [3], Jake A. Lacey [4], Karen J. Breese [5], Mark P. G. van der Linden[6], Philip M. Giffard[7,8], Steven Y. C. Tong [4,9], Nicola Gray [3], Keith A. Stubbs [5], Jonathan R. Carapetis[1,10], Asha C. Bowen[1,10], Mark R. Davies [2] & Timothy C. Barnett [1,11] ✉

Described antimicrobial resistance mechanisms enable bacteria to avoid the direct effects of antibiotics and can be monitored by in vitro susceptibility testing and genetic methods. Here we describe a mechanism of sulfamethoxazole resistance that requires a host metabolite for activity. Using a combination of in vitro evolution and metabolic rescue experiments, we identify an energy-coupling factor (ECF) transporter S component gene (*thfT*) that enables Group A *Streptococcus* to acquire extracellular reduced folate compounds. ThfT likely expands the substrate specificity of an endogenous ECF transporter to acquire reduced folate compounds directly from the host, thereby bypassing the inhibition of folate biosynthesis by sulfamethoxazole. As such, ThfT is a functional equivalent of eukaryotic folate uptake pathways that confers very high levels of resistance to sulfamethoxazole, yet remains undetectable when Group A *Streptococcus* is grown in the absence of reduced folates. Our study highlights the need to understand how antibiotic susceptibility of pathogens might function during infections to identify additional mechanisms of resistance and reduce ineffective antibiotic use and treatment failures, which in turn further contribute to the spread of antimicrobial resistance genes amongst bacterial pathogens.

Antimicrobial resistance (AMR) is a major threat to global human health that was recently estimated to directly cause 1.27 million deaths per year[1]. Genomic surveillance of AMR is one of the strategies outlined by the World Health Organization to mitigate the burden of AMR infections and guide the optimal use of existing antibiotics[2]. Surveillance is facilitated by online databases of AMR genes and mutations[3],

each of which have been defined by their ability to confer direct resistance to antibiotics in vitro[4]. These generally fall into four canonical mechanisms: antibiotic inactivation, reduced antibiotic accumulation, target expression changes, and target protection (including acquisition of functional targets with reduced antibiotic affinity)[3–5]. While new antibiotic resistance genes are continually being identified,

---

**Fig. 1 | Overview of folate metabolism in bacteria and mammals and targets of SMX and TMP.** Orange arrows are specific to bacteria. Blue arrows are pathways specific to mammals. Black arrows are common to both bacteria and mammals. Mammalian dihydrofolate reductase (mDHFR) is able to reduce folic acid to DHF, while bacterial dihydrofolate reductase enzymes are unable to perform this reaction[23]. Mammalian exogenous folate uptake pathways can accommodate multiple reduced folate substrates (RFC and hFR) or folic acid in addition to reduced folates (PCFT)[42]. RFC is the major transporter in systemic tissues while hFR and PCFT are tissue-specific. RFC reduced folate carrier, hFR human folate receptor, PCFT proton-coupled folate transporter, GTP guanosine triphosphate, DHP dihydropteroate, DHF dihydrofolate, THF tetrahydrofolate, PABA para-aminobenzoic acid, dUMP deoxyuridine monophosphate, dTMP deoxythymidine monophosphate, TMP trimethoprim, SMX sulfamethoxazole, DHFR dihydrofolate reductase (mammalian). Created with BioRender.com.

the underlying mechanisms of resistance almost exclusively fall into these four canonical mechanisms. However, existing AMR genes and mutations do not explain all antibiotic-resistant infections.

The folate biosynthesis pathway has been a target for antibiotic therapy for over 80 years[6]. The end products of this pathway, 7,8-dihydrofolate (DHF) and 5,6,7,8-tetrahydrofolate (THF), are key intermediates in the folate cycle that is required for production of deoxythymidine monophosphate. As mammals lack the enzymes required for de novo folate synthesis[7], they must acquire reduced folate cycle intermediates from dietary sources. The most common antifolate antibiotic therapy is a highly synergistic combination of two antibiotics that simultaneously inhibit de novo folate synthesis (sulfamethoxazole, SMX) and the folate cycle (trimethoprim, TMP; Fig. 1). This combination (co-trimoxazole, SXT) is commonly used to treat a variety of infections that include community-associated methicillin-resistant *Staphylococcus aureus*, Gram-negative urinary tract, and *Pneumocystis* pneumonia infections. After promising clinical trial results[8], SXT is now recommended to treat *Streptococcus pyogenes* (Group A *Streptococcus*, GAS) skin infections in endemic settings[9,10].

All described resistance mechanisms to TMP and SMX involve mutations in the target enzymes of each antibiotic (dihydrofolate reductase and dihydropteroate synthase, respectively), or horizontally-acquired enzyme variants with reduced antibiotic affinity[11]. In GAS, individual resistance to TMP and SMX has been attributed to mutation of the target enzymes (Dyr and FolP, respectively), or the acquisition of TMP-resistant variants of Dyr (e.g. DfrF and DfrG[6,12–14]). However, no horizontally-acquired SMX genes have been described in GAS and the combinations of SMX and TMP resistance that lead to SXT resistance remain unclear. In this study we investigated SXT resistance in clinical GAS isolates and identified a mechanism of SMX resistance that involves the acquisition of reduced folates from the extracellular environment. This mechanism of resistance is

mediated by a horizontally acquired energy-coupling factor (ECF) transporter S component (ThfT) that expands the substrate profile of an endogenous ECF transporter to include reduced folate cycle intermediates. GAS strains that encode ThfT are highly resistant to SMX in the presence of reduced folates yet remain sensitive when grown in the absence of these compounds. As a result, ThfT-positive GAS strains remain sensitive to SMX when grown under laboratory conditions that are routinely used to monitor antibiotic susceptibility, yet are likely resistant to SMX in vivo as a result of acquisition of reduced folates directly from the host.

## Results

### In vitro resistance of GAS to SXT requires high-level TMP resistance conferred by *dfrF*

To investigate the combinations of TMP and SMX resistance that confer SXT resistance, we determined the susceptibility for SMX, TMP and SXT for a global collection of GAS strains reported to have reduced susceptibility to SXT[13,15] (Supplementary Table 1). The minimal inhibitory concentration (MIC) for each strain was determined using Epsilometer test (Etest) strips and Mueller-Hinton Fastidious agar (MHF; BioMerieux). While each strain was originally reported as being SXT-resistant using broth microdilution methodology[13,15], only 5/22 were SXT-resistant using Etest methodology. This discrepancy is potentially explained by methodological differences between these assays, or differences in the composition of the growth medium used in each assay.

To identify genetic determinants associated with SXT resistance, we next compared MIC values for each antibiotic with known antibiotic resistance genes identified by whole genome sequencing (Supplementary Table 1). Supporting previous reports[12,13], we identified TMP-resistance genes *dfrF* and *dfrG* in GAS isolates that exhibited TMP MIC values >32 μg/ml. *dfrF* and *dfrG* were variably encoded on horizontally acquired genetic elements (e.g. integrative conjugative elements), often with a number of additional antibiotic resistance genes. While several strains were found to have SMX MICs >100 μg/ml that were sufficient for SXT resistance, we were unable to identify any horizontally-acquired dihydropteroate synthase variants that would account for SMX resistance in these strains. Inspection of the sequence of the SMX target enzyme (FolP) in each strain revealed considerable sequence variation, yet there were no individual amino acid sequence variations that correlated with reduced SMX susceptibility (Supplementary Fig. 1). Structural[16] and biochemical[17] investigations have identified key residues in dihydropteroate synthase enzymes involved in SMX drug binding and resistance, which correspond to residues F25, S26, T59 and P61 in the GAS FolP sequence (Supplementary Fig. 1). Of these, only T59 showed any variation between the strains investigated, however the two amino acid variations at this residue did not correlate with the observed SMX MICs for strains that contained them (T59A, MIC range 48 to >1024 μg/ml; T59N, MIC 48 μg/ml). Thus, while we can not discount that combinations of FolP amino acid changes might have contributed to a reduced SMX susceptibility in some of these strains, the exact mechanism of resistance remained unclear.

### *dfrF* confers much higher levels of TMP resistance than *dfrG*

Comparison of SXT susceptibility results for each strain in Supplementary Table 1 revealed that only *dfrF*-positive strains were SXT-resistant (MIC > 2 μg/ml). Surprisingly, both *dfrG*-positive strains (TB08 and TB10) were highly-resistant to TMP (MIC > 32 μg/ml) and SMX (MICs > 1024 μg/ml) yet remained SXT-sensitive. As TMP is a competitive inhibitor of substrate binding to dihydrofolate reductase enzymes, resistant enzyme variants have reduced affinity for TMP such that higher levels of antibiotic are required to inhibit enzyme activity[18]. As each *dfrF*- and *dfrG*-positive GAS strain had TMP MICs greater than the highest level resolved with the TMP Etest (32 μg/ml), we next compared levels of TMP susceptibility using a gradient plate assay

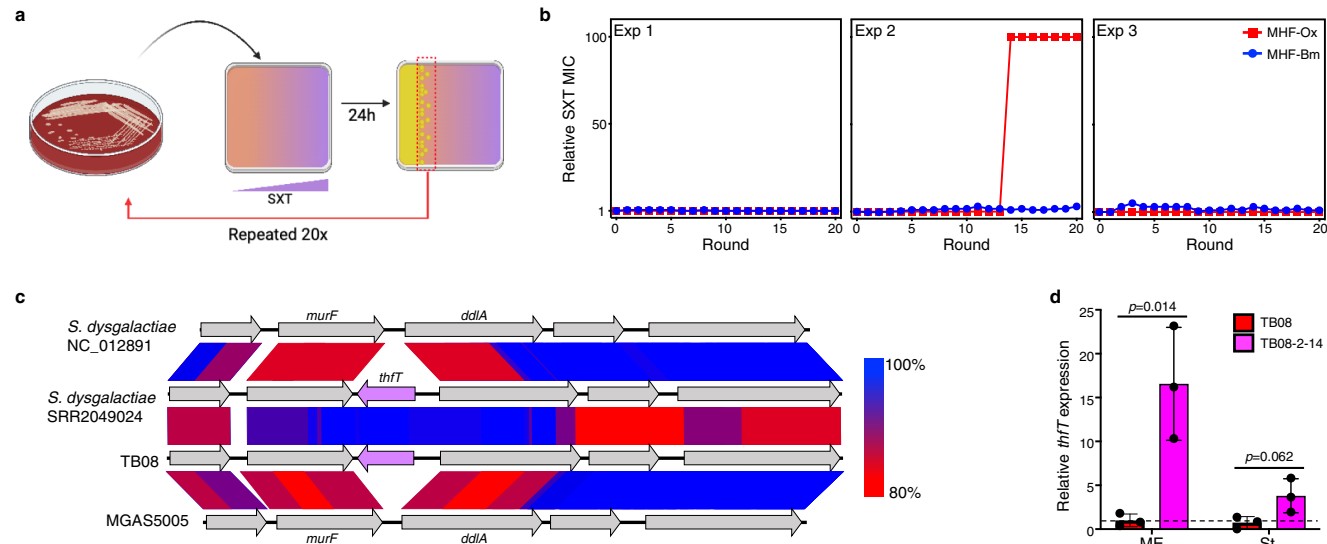

**Fig. 2 | A DfrG-positive GAS strain can evolve SXT resistance due to upregulation of an ECF transporter S component gene (*thfT*). a** Schematic of in vitro evolution procedure using SXT gradient plates. An overnight culture of TB08 on HBA was used to inoculate a SXT gradient plate (0–50 μg/ml) and incubated for 24 h at 37 °C. Growth was then harvested from the the highest antibiotic concentration (i.e. where growth started to be inhibited) and used to seed successive rounds of evolution. An overview of this process is illustrated in Fig. 2a. Created with BioRender.com. **b** MICs to SXT on MHF agar from bioMerieux (MHF-Bm; blue circles) and Oxoid (MHF-Ox; red squares) following in vitro evolution of GAS strain TB08. Results from three independent experiments are shown. Values are expressed as a ratio of the initial MIC as determined by Etest following each round of evolution. **c** Genomic context of *thfT* (purple arrow) as determined by tblastX pairwise blast which highlights evidence of horizontal gene transfer of *thfT* (purple) and shared origin with that in *Streptococcus dysgalactiae* subsp *equisimilis*. Percent identity is by shading as indicated to the right. **d** Relative expression of *thfT* in TB08 and TB08-2-14 measured by RT-qPCR. Values are normalised to average expression of TB08 ME samples (dashed line). ME, midexponential phase ($OD_{600}$ = 0.4–0.6); St, stationary phase ($OD_{600}$ = 1.2–1.4). Data from three biological replicates are presented as mean values ± SD. Differences assessed using a one-tailed, unpaired Student's *t*-test. Source data for panels **b** and **d** are provided as a Source Data file.

(Supplementary Fig. 2A). Both *dfrG*-positive GAS strains were resistant to TMP concentrations between 50 and 200 μg/ml, while each of the *dfrF*-positive strains were resistant to more than 1000 μg/ml. Thus, the SXT-resistant phenotype of *dfrF*-positive GAS is likely attributed to their very high levels of TMP resistance, while the SXT-sensitive phenotype of *dfrG*-positive GAS is likely attributed to the lower levels of TMP resistance combined with highly synergistic activity of SMX and TMP[19] (Supplementary Fig. 2B).

### Forced evolution of *dfrG*-positive GAS strain TB08 leads to SXT resistance through upregulation of a ECF transporter substrate-binding component (*thfT*)

While *dfrG*-positive GAS isolates were sensitive to SXT, we reasoned that they might have the capacity to evolve SXT resistance with strong antibiotic selection. To investigate this, we examined the ability of a *dfrG*-positive/SMX-resistant/SXT-sensitive strain (TB08) to evolve SXT resistance following 20 rounds of serial passage on inhibitory levels of SXT. To do this, a single TB08 colony was spread over the surface of an agar plate containing a gradient of SXT (0–50 μg/ml) and cultured overnight. After each round of growth, colonies that grew at the highest concentration of SXT were used to seed subsequent rounds of evolution. An overview of this process is illustrated in Fig. 2a. After each round of selection, SXT susceptibility was monitored on SXT gradient plates (Supplementary Fig. 3) and by Etest (Fig. 2b). In one of three biological replicates, we observed a switch of TB08 to a SXT-resistant phenotype with the gradient plate assay (Supplementary Fig. 3), however a comparable SXT-resistant switch was not observed by Etest (Fig. 2b, blue circles).

In the in vitro evolution experiments above, the gradient plate (Supplementary Fig. 3) and Etest (Fig. 2b) assays used to monitor SXT susceptibility used different formulations of growth media. Gradient plates were prepared using Mueller-Hinton (MH) agar base (Oxoid) containing 2.5% lysed horse blood, while Etest assays were conducted on commercial Mueller-Hinton Fastidious agar plates (Mueller-Hinton

agar containing 5% defibrinated horse blood and 20 μg/ml β-NAD from BioMerieux; MHF-Bm). To examine whether differences in media composition accounted for the differences in SXT MICs between the gradient plate and Etest assays, we next measured SXT MICs using Etest strips on MHF prepared using MH agar base from Oxoid (MHF-Ox; Fig. 2b, red squares). This analysis showed that the observed switch of TB08 to a SXT-resistant phenotype was dependent on the media composition, and only observed on MHF-Ox. Further examination of the TB08 parent and the evolved TB08 (from Experiment 2, round 14; strain TB08-2-14) revealed that TB08 and TB08-2-14 remained SXT-sensitive on MHF-Bm, while TB08-2-14 was resistant on MH agar from Oxoid containing different combinations of horse blood, lysed horse blood and β-NAD (Supplementary Fig. 4). These results demonstrate that the *dfrG*-positive/SMX-resistant isolate TB08 can switch to a SXT-resistant phenotype under strong selective pressure, and detection of this switch depends on the composition of the growth medium.

To identify the mechanism of evolved resistance in TB08-2-14, we performed whole genome sequencing on this strain and compared this sequence with the parent strain TB08. This comparison revealed that the SXT-resistant phenotype of TB08-2-14 was associated with a 5 bp sequence duplication in the upstream intergenic region of a putative folate transporter gene (herein named *thfT*). *thfT* encodes a protein with homology to FolT, the folate-specific substrate-binding component (S component) of an energy-coupling factor (ECF) transporter[20,21] from *Lacticaseibacillus casei*[22] (28% identity, 46% similarity) and *Leuconostoc mesenteroides*[20] (23% identity, 45% similarity; Supplementary Fig. 5). Comparison with other SXT susceptible GAS genomes (e.g. MGAS5005, Fig. 2c), revealed *thfT* is inserted between *murF* and *ddlA*, two genes that comprise part of the core GAS genome and encode enzymes involved in peptidoglycan biosynthesis.

The 5 bp duplication in TB08-2-14 is in the intergenic region between *thfT* and the divergently transcribed gene *ddlA*. As this intergenic region likely contains the promoter region for *thfT*, we next examined whether the increased SXT resistance in TB08-2-14 was

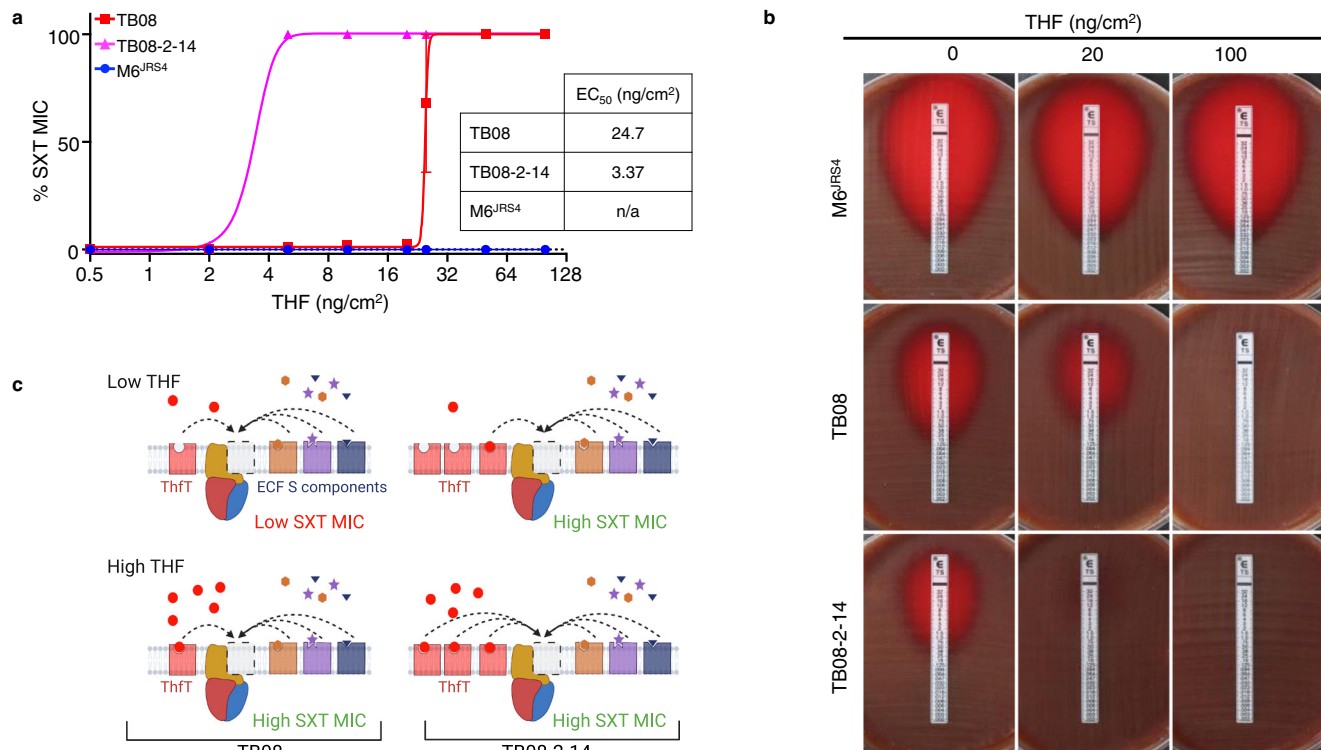

**Fig. 3 | ThfT confers SXT resistance in the presence of exogenous THF. a** Dose-response curves for the impact of exogenous THF on SXT resistance of M6[JRS4], TB08 and TB08-2-14. MICs are expressed as a percentage the maximum resolution of the Etest assay (32 µg/ml). Data from three biological replicates on MHF-Bm agar are presented as mean values ± SD. EC$_{50}$ values were calculated using Graphpad Prism software. **b** Representative Etest assays on MHF-Bm agar for data presented in **a**. **c** Schematic showing predicted effect of increased ThfT expression in TB08-2-14 on competition with other ECF S components for ECF transport modules and the effect on SXT MICs. Under conditions of low ThfT expression and low exogenous THF, competition between THF-loaded ThfT and S components loaded with other compounds limits THF import, resulting in low SXT MICs (top left). This competition is overcome by increased ThfT expression (top right) or increased extracellular THF, which results in high SXT MICs. Source data for panel **a** is provided as a Source Data file.

associated with upregulation of the *thfT* gene. RNA was prepared from each strain and *thfT* mRNA levels were quantified by qRT-PCR (Fig. 2d). Results from this experiment demonstrate that the 5 bp duplication in TB08-2-14 is associated with an increased level of *thfT* transcript in both mid-exponential (16.6-fold) and stationary (3.8-fold) phases of growth relative to TB08. This suggests that *thfT* could be an AMR gene that is up-regulated in TB08-2-14 due to a mutation in the *thfT* promoter region.

## ThfT enables GAS to bypass the inhibitory effects of SXT through acquisition of extracellular tetrahydrofolate (THF)

The above results suggest that increased expression to *thfT* in TB08-2-14 is responsible for the increased SXT resistance of this strain relative to the parent strain TB08. Given the homology of ThfT to the substrate binding S component of an ECF folate transporter, we hypothesised that ThfT might expand the substrate profile of an endogenous ECF transporter to include end products of the folate biosynthesis pathway. This would enable *thfT*-positive GAS strains to bypass the inhibitory effects of SMX and/or TMP (Fig. 1). To examine this possibility, we first tested SXT MICs for TB08 and TB08-2-14 and a *thfT*-negative strain (M6[JRS4]) in the presence of different amounts of THF (Fig. 3a, b; Supplementary Fig. 6) and folic acid (Supplementary Fig. 7). While the *thfT*-negative strain M6[JRS4] did not exhibit any change in SXT MIC in the presence of either compound, both TB08 and TB08-2-14 showed an increased SXT MIC in the presence of extracellular THF but not folic acid, with both strains being resistant to >32 µg/ml SXT at THF concentrations >50 ng/cm² THF. The lack of resistance in the presence of extracellular folic acid likely reflects that folic acid is a very poor substrate for most bacterial dihydrofolate reductase enzymes[23]. A dose-response comparison by Etest showed that the amount of THF

required to increase SXT resistance in the evolved strain TB08-2-14 (EC$_{50}$ = 3.37 ng/cm²; Fig. 3a) was lower than the amount required for the parent strain TB08 (EC$_{50}$ = 24.7 ng/cm²; Fig. 3a). Similar results were obtained by broth microdilution assay (TB08-2-14 EC$_{50}$ = 1.72 ng/ml, TB08 EC$_{50}$ = 7.60 ng/ml; Supplementary Fig. 6). These results are likely explained by the increased levels of ThfT in TB08-2-14, which may then outcompete other ECF S components for the same ECF transport module (Fig. 3c). This is supported by recent evidence that ECF S components can compete for the same ECF transport module, with high levels of substrate for one S component (e.g. pantothenate) able to reduce uptake of second substrate (e.g. folate) that is transported by an alternative S component[24]. Together, these data indicate that THF or related compounds are likely present in MH agar sourced from Oxoid at an intermediate concentration sufficient to support SXT resistance for TB08-2-14 but not for TB08 (Fig. 3a). In contrast, these compound(s) are absent or at much lower concentrations in MHF agar from BioMerieux (MHF-Bm) and are unable to support SXT resistance for either strain.

## ThfT is a SMX resistance protein that confers very high levels of SMX resistance

Comparative genomics of 24 globally sourced GAS strains (Supplementary Table 1) revealed five strains that encoded *thfT*. To investigate whether all strains that encoded *thfT* were SXT resistant with exogenous THF, we next measured SXT MICs of each strain in the absence and presence of 50 ng/cm² THF (Supplementary Table 2). Results showed that each of the *thfT*-positive strains that also encoded a TMP resistance gene (*dfrG* or *dfrF*) and were TMP resistant (>32 µg/ml) became highly-resistant to SXT in the presence of exogenous THF, while both of the strains that lacked a TMP resistance gene exhibited a

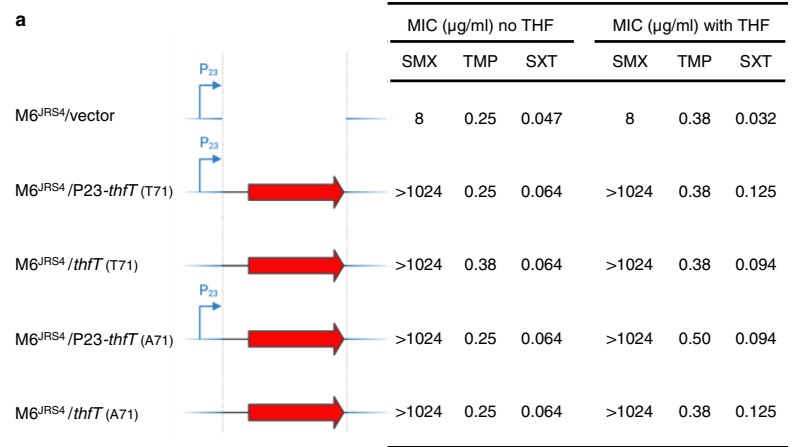

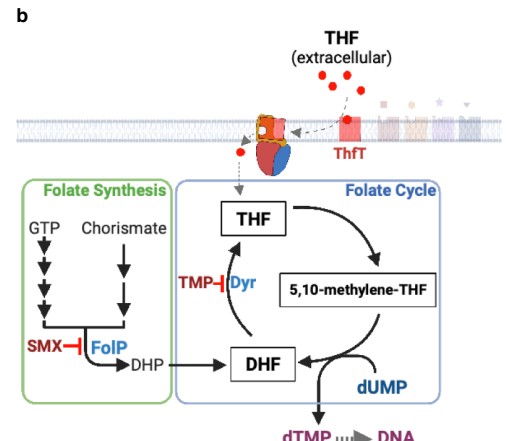

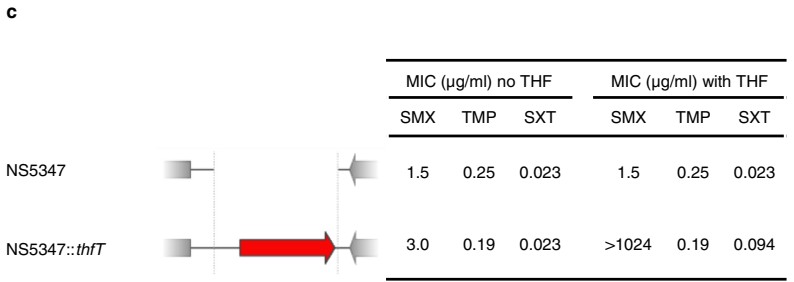

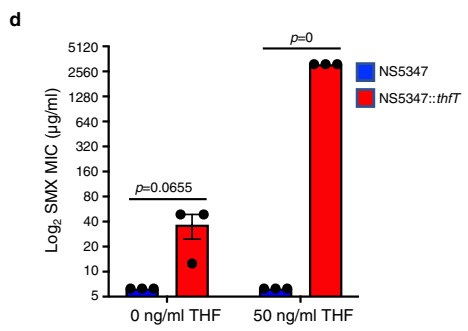

**Fig. 4 | ThfT is an SMX resistance protein that allows GAS strains to bypass the folate synthesis pathway in the presence of exogenous THF. a** Heterologous expression of T71 and A71 variants of *thfT* in M6^JRS4^. The constructs with a constitutive P23 promoter are indicated with a blue arrow and *thfT* is indicated with a red arrow. SMX, TMP and SXT MICs were determined by Etest. Exogenous THF was supplied at a concentration of 50 ng/cm². Results are median values of three independent experiments determined with Etest strips on MHF-Bm agar. Statistical analysis of groups is in Supplementary Table 4. **b** Model of ThfT mediated SMX-resistance. ThfT is able to acquire extracellular THF which then feeds directly into the folate cycle (blue box), which bypasses the inhibitory effect of SMX on folate synthesis (green box). As THF is recycled back to DHF during the folate cycle, only small amounts of exogenous THF are required to support growth in the presence of SMX. GTP, guanosine triphosphate; DHP, dihydropteroate; DHF, dihydrofolate; dUMP, deoxyuridine monophosphate; dTMP, deoxythymidine monophosphate. Created with BioRender.com. **c** Single gene complementation of *thfT* in the same genetic locus as in TB08 is sufficient for high-level SMX resistance in GAS strain NS5347. SMX, TMP and SXT MICs were determined by Etest. Exogenous THF was supplied at a concentration of 50 ng/cm². Results are median values of three independent experiments determined with Etest strips on MHF-Bm agar. Statistical analysis of groups is in Supplementary Table 5. **d** Ability of THF to confer ThfT-mediated SMX resistance as measured by broth microdilution. Data from three biological replicates are presented as mean values ± SEM. Differences assessed using a two-tailed, unpaired Student's *t*-test. Source data for panels **a**, **c** and **d** are provided as a Source Data file.

modest (~4 to 5-fold) increase in SXT MIC values in the presence of THF.

Alignment of the predicted ThfT amino acid sequences identified two allelic variants that differed by a single amino acid (T71 and A71). Each of the strains with the T71 variant also encoded a TMP resistance gene, while both strains with the A71 variant lacked a TMP resistance gene and were TMP sensitive. To rule out any effects in affinity for THF in the A71 variant that might explain the different phenotype of the strains that encoded each variant, we examined SXT resistance across a range of THF concentrations (Supplementary Table 3). Addition of increasing extracellular THF (up to 400 ng/cm²) was unable to confer high-level SXT resistance for either of the A71 variant strains, suggesting that the lower level of SXT resistance in the A71 strains was not due to a reduced affinity for THF. Together, these results suggest that ThfT may be functioning as a SMX resistance protein, and that ThfT-mediated SXT resistance requires an additional mechanism of TMP resistance.

To further investigate whether ThfT conferred SXT resistance or SMX resistance, we expressed each ThfT variant in a laboratory GAS strain (M6^JRS4^) that lacked any TMP- or SMX-resistance mechanism, and measured the effect of exogenous THF on MIC values for SMX, TMP

and SXT (Fig. 4a, Supplementary Table 4). Expression of either variant resulted in a very modest increase in SXT MIC in the presence of 50 ng/cm² of exogenous THF, yet had no effect on MIC values obtained for TMP. However, expression of either variant resulted in high level SMX resistance (>1024 µg/ml) even in the absence of additional extracellular THF. This result was unexpected since addition of exogenous THF is required for SXT resistance in the parent strain (Fig. 3a, b). We rationalised that the lack of a requirement for additional THF for SMX resistance in this experiment may be explained by higher expression levels of *thfT* when expressed on a multicopy plasmid, which may then require lower levels of exogenous THF to bypass SMX inhibition of the folate synthesis pathway. The requirement of only small amounts of extracellular THF is likely explained by the fact that THF is recycled back to DHF during the production of thymidine monophosphate, such that only very low levels of THF should theoretically support DNA synthesis (Fig. 4b).

To further investigate the requirement of extracellular THF for ThfT-mediated SMX resistance and control for any experimental artefacts caused by gene dosage and strain background, we next inserted the *thfT* T71 variant into the genome of a *thfT*-negative GAS strain by homologous recombination. For this experiment, we used

NS5347 as the parent strain as it belonged to the same multi-locus sequence type (MLST 289) as TB08 and had the same nucleotide sequences flanking the *thfT* locus. This enabled efficient recombination of *thfT* into exactly the same locus as in TB08. Examination of each strain in the absence and presence of 50 ng/cm$^2$ exogenous THF revealed that *thfT* was able to confer high levels of SMX resistance (>1024 μg/ml), but only in the presence of exogenous THF (Fig. 4c, Supplementary Table 5). This resulted in a very modest increase in SXT MIC (~4.5-fold) with no associated increase in TMP MIC. These results demonstrate that ThfT functions as a SMX resistance determinant that imports extracellular THF and bypasses the inhibitory activity of SMX on the folate biosynthesis pathway (Fig. 4b).

If ThfT allows GAS to bypass the folate synthesis pathway altogether in the presence of extracellular THF, then strains that harbor *thfT* should have complete resistance to SMX in the presence of extracellular THF. As the maximum resolution of the Etest is 1024 μg/ml, we compared relative SMX resistance levels of NS5347 and NS5347::*thfT* by broth microdilution across a range of SMX concentrations up to 5 mg/ml (the maximum concentration we could dissolve into the growth media) in the absence and presence of THF. Using this method, we found that NS5347::*thfT* was able to grow in the presence of at least 5 mg/ml of SMX in the presence of exogenous THF but not in the absence of exogenous THF (Fig. 4d). Taken together, these results demonstrate that *thfT* encodes an antibiotic resistance protein that provides near-absolute resistance to SMX by bypassing the direct inhibitory action of this antibiotic on the folate synthesis pathway.

### ThfT enables GAS to acquire multiple one-carbon THF molecules from host cells

In addition to the role of THF as an intermediate in the folate cycle, THF is also used as a carrier for one-carbon molecules in biosynthetic reactions, which are ultimately recycled back to THF[7]. The differences in the susceptibility of TB08-2-14 to SXT on MHF-Ox and MHF-Bm media suggest that there are differences in the concentration of THF or related compounds in these media. To investigate this, we compared the composition of MH base medium from each supplier using nuclear magnetic resonance (NMR) spectroscopy and reversed phase LC MS/MS approaches (Supplementary Note 1). While we found major differences in the composition of MH base medium from each supplier by both methods, we were unable to identify substantial differences in folate pathway compounds that might explain the relative performance of each media for detecting ThfT-mediated SMX resistance. We next examined the ability of different reduced folate compounds to directly rescue SMX resistance in GAS expressing ThfT. For this experiment, we used the NS5347 strain derivatives encoding the *thfT* T71 variant inserted as a single gene copy into the chromosome (NS5347::*thfT*). SMX resistance of each strain was determined by Etest in the presence of 50 ng/cm$^2$ of each different exogenous one-carbon THF compound spread over the surface of the agar plate (Fig. 5a; Supplementary Tables 6 and 7). Expression of ThfT conferred very high SMX resistance to NS5347 in the presence of all one-carbon THF intermediates except for 5-methyl-THF, likely because GAS strains lack homologues of the enzymes required to convert 5-methyl-THF back to THF (Fig. 5b; Supplementary Fig. 8). These results demonstrate that the ThfT T71 variant is able to import multiple one-carbon THF compounds from the extracellular environment.

The above results show that ThfT enables uptake of multiple reduced folate compounds that comprise the intracellular folate pool in host cells. We next tested the ability of ThfT to acquire reduced folates directly from primary human epithelial cells. For this experiment, we used a modified broth microdilution assay to allow measurement of SMX MICs in the presence of different concentrations of epithelial cell lysate (Fig. 5c). Results of this experiment show that the cellular folate pool is sufficient to rescue growth of NS5347::*thfT* but not the NS5347 parental strain (Fig. 5d). Thus, ThfT is able to acquire reduced folates from host cells and bypass SMX inhibition of folate synthesis. We propose that cellular folate cycle intermediates and one-carbon THF pools serve as multiple in vivo substrates for ThfT-mediated SMX resistance (Fig. 5b), which are likely released during infection from neutrophils recruited to the site of infection and/or tissue damage that occurs due to production of one or more GAS cytolysins[25,26].

### *thfT* is likely acquired by horizontal transfer from related *Streptococcus* species

Supplementary Table 1 shows that *thfT* was variably encoded by the GAS strains examined in this study. To obtain a more comprehensive view on *thfT* carriage and SXT resistance in GAS strains, we investigated the presence of *thfT*, *dfrF* and *dfrG* in a global dataset of 2083 diverse GAS isolates[27] (Supplementary Table 8). Carriage of *thfT* was uncommon in this dataset (20/2083 GAS strains) yet maintained high levels of sequence homology. Furthermore, *thfT*-positive strains included 11 unique *emm* sequence types that were from multiple geographic regions. Of these, 6/20 *thfT*-positive strains also encoded a TMP resistance gene (*dfrF* or *dfrG*; Supplementary Table 8) and are likely functionally SXT resistant in the presence of extracellular reduced folate compounds.

A gene identical to *thfT* (annotated as *folT*) was recently described to be carried on a similar genetic element in a subset of GAS, *S. dysgalactiae* subsp. *equisimilis* and *S. dysgalactiae* subsp. *dysgalactiae* strains[28], suggesting a likely common origin. Examination of publicly available *S. dysgalactiae* genome sequences from refseq revealed that carriage of *thfT* was also uncommon in this species (3/136 strains; Supplementary Table 9), with no strains identified as carrying both *thfT* and a TMP resistance gene.

## Discussion

Antimicrobial resistance (AMR) is defined as the inherited ability of a bacterial strain to grow in high antibiotic concentrations as measured by a high minimal inhibitory concentration (MIC)[29]. The entire field has largely been defined by bacterial growth on laboratory media and each mechanism enables the resistant organism to grow in the presence of the antibiotic in vitro. In this study, we identified a mechanism of AMR mediated by ThfT that works by expanding the substrate profile of an endogenous ECF transporter to include the end products of the folate synthesis pathway. Through the acquisition of extracellular reduced folates, ThfT enables GAS to bypass the inhibition of folate synthesis by SMX altogether. This provides very high levels of resistance to SMX, which enables a lower level of TMP resistance (such as that provided by DfrG) to be sufficient for SXT resistance (Supplementary Fig. 9). As reduced folates are abundant in host cells, ThfT-mediated resistance is likely functional in the context of a host infection yet remains undetected on routine laboratory testing media. As such, there is potential for strains that harbour *thfT* to emerge and disseminate undetected by current AMR surveillance methods. We propose that ThfT-mediated SMX resistance is a host-dependent AMR mechanism (Supplementary Fig. 10), and our findings provide the opportunity to monitor any emergence of *thfT*-positive GAS and treat these infections with alternative antibiotics.

Changes in core metabolism have recently been identified as an important process by which bacteria can acquire antibiotic resistance. This is evidenced by the selection of mutations in core metabolism genes in *Escherichia coli* that lower metabolic activity and reduce antibiotic lethality[5]. Our discovery of ThfT provides a mechanism for changes in nutrient availability to also mediate AMR, and is an example of a horizontally-acquired AMR mechanism that involves the uptake of normally inaccessible nutrients. The identification of a nutrient uptake system mediating this resistance mechanism also provides an

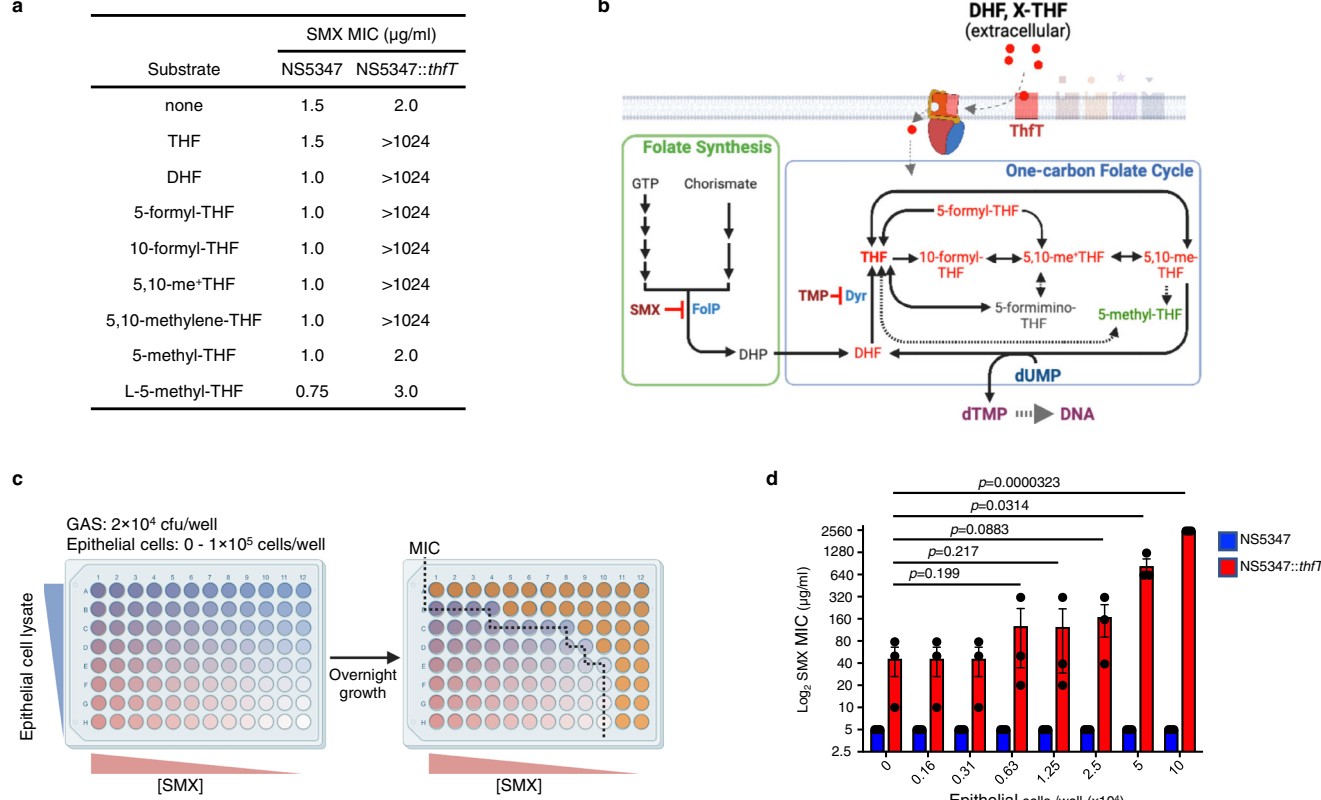

**Fig. 5 | ThfT enables SMX-resistance by uptake of multiple one-carbon folate cycle intermediates from host cells. a** Ability of NS5347 encoding *thfT* variants to utilise multiple one carbon THF intermediate compounds. SMX MICs (µg/ml) were determined by Etest in the presence of 50 ng/cm² of each compound on MHF-Bm agar. Results are median values from three independent experiments. Statistical analysis of groups is in Supplementary Table 7. 5,10-me⁺THF, 5,10-methenyl-THF. **b** Model of ThfT mediated SMX resistance in the presence of one-carbon folate cycle intermediate compounds. Compounds utilised by ThfT to bypass the inhibitory action of SMX are highlighted in red and compounds which could not be utilised by ThfT are highlighted in green. 5-formimino-THF (grey) was not available for testing. Solid black arrows indicate pathways that are likely functional in GAS based on the presence of predicted enzymes in the core GAS genome, while pathways indicated with dashed grey arrows indicate the absence of corresponding

enzyme[43] (Supplementary Fig. 8). Created with BioRender.com. GTP guanosine triphosphate, DHP dihydropteroate, DHF dihydrofolate, dUMP deoxyuridine monophosphate, dTMP deoxythymidine monophosphate, 5,10-me-THF 5,10-methylene-THF. **c** Cartoon schematic of experiment to test uptake of cellular folates by ThfT. SMX MICs were by broth microdilution across a 2-fold serial dilution of SMX (2500 to 5 µg/ml, left to right) in the presence of a 2-fold serial dilution of epithelial cell lysates (1 × 10⁵ to 1.6 × 10³ cells/well, top to bottom). After overnight incubation, MIC values were determined by examining GAS growth (orange wells) for each epithelial cell lysate concentration. **d** Ability of epithelial cell lysates to confer ThfT-mediated SMX resistance using procedure in **c**. Data from three biological replicates are presented as mean values ± SEM. Differences assessed using a one-tailed, paired Student's *t*-test. Source data for panels **a** and **d** are provided as a Source Data file.

interesting layer of complexity in monitoring AMR, since the choice of bacterial strain and growth conditions could vastly affect the susceptibility profile to different antibiotics depending on what nutrients are available in the growth medium and whether the bacterial strain is able to take up these nutrients. This may explain why Mueller-Hinton media sourced from different suppliers can have vastly different effects on the susceptibility of bacteria to different antibiotics, including to antibiotics that do not target metabolic pathways[30]. Our results support this study. Furthermore, the vastly different nutrient profiles in media from different suppliers likely underpins these observations. The different nutrient profiles of MH media from two reputable suppliers (Supplementary Note 1) was a surprising finding given that there are published standards and performance criteria for the manufacture of MH agar[30]. We suggest that efforts be made to understand the effect of these differences on antibiotic susceptibility for different pathogens[30], as they may affect surveillance of AMR and may lead to the discovery of new AMR mechanisms that are mediated by nutrient availability.

While the combinations of TMP and SMX resistance genes required for SXT resistance are currently rare in global GAS strains, the apparent transmissibility of these genes means that emergence of SXT-resistant GAS should be monitored with increasing use of this

antibiotic to treat GAS skin infections. Both *dfrG*[12] and *dfrF*[31] have been found on genetic elements that are likely transferrable between GAS strains and related *Streptococcus* species, with *dfrF* located on integrative-conjugative elements along with multiple additional AMR genes. This raises the concern that *dfrF* could not only be selected for by increased use of SXT, but also with additional antibiotics. We also provide evidence that *thfT* is transferred horizontally between GAS strains and related *Streptococcus* species. This is based on the fact that *thfT* is encoded by multiple GAS *emm* sequence types, yet not universally encoded by GAS strains belonging to each of these individual types or sequence types (e.g. TB08 and NS5347 both belong to *emmST* 4.2 and MLST 289 yet differ in their carriage of *thfT*). Our results are supported by a recent study[28], and suggest horizontal gene transfer of *thfT* between related *Streptococcus* species is the primary method of acquisition of this element. Intriguingly, the *thfT* locus is also adjacent to a horizontally-acquired penicillin-binding protein (*pbp2b*)[28]. While the significance of *pbp2b* carriage by GAS on penicillin susceptibility remains to be determined, if *pbp2b* does reduce the susceptibility of GAS to penicillin then selection for *thfT* acquisition might also occur during penicillin treatment. Combined with our functional characterization of ThfT, this suggests that *thfT* is a horizontally acquired AMR gene that is able to transfer between related *Streptococcus* species. Our

results therefore provide the opportunity to monitor any such emergence of resistance to SXT and modify antibiotic therapy accordingly.

AMR is a global healthcare emergency that requires action on multiple fronts, including new antibiotic discovery[32], surveillance of resistance[33], and understanding additional mechanisms of antibiotic failure such as tolerance and persistence[29]. AMR surveillance has traditionally involved routine testing on laboratory media and detection of AMR genes through whole genome sequencing, however, existing AMR mechanisms do not explain all antibiotic treatment failures. Our findings highlight the need to understand antibiotic activity in the context of the infections they are designed to treat, and should serve as a paradigm for investigating additional mechanisms of host-dependent AMR in medically-important bacterial pathogens. In order to preserve the long-term efficacy of antibiotics, we need to identify and understand these additional mechanisms of antibiotic treatment failures to aid in the discovery of new antibiotics and monitor AMR as it arises.

## Methods

### Bacterial isolates, media and growth conditions
GAS strains used in this study are listed in Supplementary Table 1. Strains were cultured at 37 °C on horse blood agar (HBA; Pathwest P018) or Todd-Hewitt broth (Bacto 249240) containing 1% yeast extract (Thermo Fisher FSBBP9727) and 1.5% bacteriological agar (Becton Dickinson 214530; THYA). Liquid cultures were prepared in Todd-Hewitt broth containing 1% yeast extract (THYB) and cultured statically at 37 °C. *Escherichia coli* strain NEB®5-alpha (C2987, New England Biolabs) was used as a host for cloning experiments and cultured in Luria–Bertani medium (LB; Thermo Fisher 22700025 and 12780052). As required, spectinomycin (Sigma Aldrich S4014) was used at 100 µg/ml for both GAS and *E. coli*.

### Antibiotic susceptibility testing
Etest assays were performed on Mueller-Hinton Fastidious agar (MHF; bioMerieux 43901) or on MHF agar prepared from Mueller-Hinton broth (Oxoid CM04050B) according to EUCAST guidelines (https://www.eucast.org/ast_of_bacteria/media_preparation/). For some experiments (as indicated), MHF agar and variations thereof were prepared from Mueller-Hinton broth (Oxoid, CM04050B) and addition of defibrinated horse blood (Serum Australis HD100D), lysed horse blood, and/or *β*- nicotinamide adenine dinucleotide (*β*-NAD; Sigma N0632). Single colonies from overnight growth of each strain on HBA were resuspended in 0.9% w/v NaCl and adjusted to 0.5 McFarland standard ($OD_{600} = 0.1$). A sterile swab was saturated with this suspension and used to inoculate the surface of an agar plate to create a confluent lawn inoculum. A single Etest strip was applied to the surface of the inoculated plate and incubated for 18 h at 37 °C. Etests were purchased from BioMerieux for SXT (412481) and TMP (523618). SMX Etests were from BioMerieux (412458) or Integrated Sciences (920311). The MIC was recorded where the zone of inhibition intersected the Etest strip scale. For SMX, TMP and SXT where the zone of inhibition appears hazy, MICs were recorded where ~80% inhibition of growth was observed. Control GAS strains (5448 and/or JRS4) were included during every round of testing. For TMP and SXT, EUCAST breakpoints were used (https://eucast.org/clinical_breakpoints/). There are no published breakpoints for SMX.

Broth microdilution tests were performed by a modification of the method described by Wiegland et al.[34] in MH-Bm broth. Overnight cultures of each bacterial strain were adjusted to 0.5 McFarland standard ($OD_{600} = 0.1$), then serially diluted to a $10^{-2}$ dilution and 100 µl added to a 96-well plate (~$2 \times 10^4$ cfu/well). Antibiotics were serially diluted two-fold and added 1:1 to the bacterial suspension. Plates were incubated for 24 h at 37 °C. MIC was recorded as the lowest concentration of each antibiotic that inhibited >80% growth.

For cell lysate experiments, primary tonsil epithelial (TEpi) cells (Sciencell 2560) were cultured in Tonsil Epithelial Cell medium (Sciencell 2561) on poly-L-lysine-coated culture vessels according to the supplier's instructions (https://www.sciencellonline.com/PS/2560.pdf)[35]. At 90–100% confluency, $7 \times 10^6$ cells were disassociated with 0.05% (*v/v*) Trypsin-EDTA (Thermo Fisher 25300062), washed then lysed in ultra-pure water followed by passing through a 26-gauge needle five times. Two-times concentrated MH-Bm broth was added for a final concentration of $5 \times 10^5$ lysed TEpi cells per mL in MH-Bm broth containing 5% lysed horse blood and used for broth microdilution experiments as described above.

### Plate gradient assay (PGA)
The PGA for TMP and SXT was an adaptation of the vancomycin gradient assay[36]. Antibiotics used were TMP (Sigma Aldrich T7883), SMX (Sigma Aldrich S7507) or a 1:19 ratio of TMP:SMX. To prepare gradient plates, 50 ml of MH agar with 2.5% lysed horse blood supplemented with desired maximum concentration of antibiotic was added on to a 10 cm square Petri plate, with one side elevated at 5° to form a two-dimensional gradient of antibiotic, and allowed to set at room temperature for 20 min. Each plate was then placed flat and second layer of 50 ml of MH agar with 2.5% lysed horse blood without antibiotics was set over the first layer. Plates were stored at 4 °C overnight to allow the complete diffusion of antibiotic to create an antibiotic gradient. For testing relative MICs, single colonies from overnight growth of each strain on HBA were resuspended in 0.9% w/v NaCl and adjusted to an $OD_{600} = 0.5$ to 0.6. A sterile swab was saturated with this suspension and used to streak the strain across the gradient plate from lowest to highest antibiotic concentration. Plates were incubated 18–24 h at 37 °C.

The procedure for SXT evolution experiments is outlined in Fig. 2a. First, a single TB08 colony was subcultured on HBA and incubated at 37 °C for 18–24 h. A suspension of overnight growth was prepared in 0.9% w/v NaCl ($OD_{600} = 0.5$ to 0.6) and was spread over an entire PGA plate (SXT 0–50 µg/ml), and incubated at 37 °C for 18–24 h. GAS growth at the highest antibiotic concentration (i.e. where growth started to be inhibited) was collected and cultured overnight on HBA at 37 °C. Overnight growth was resuspended in 0.9% w/v NaCl, and used to measure MIC (by Etest, as above) and to seed the subsequent rounds of evolution as above. This procedure was repeated for 20 rounds of evolution and three independent experiments.

### Whole genome sequencing
Genomic DNA was prepared from overnight 1.8-ml THYB cultures of each GAS strain using the DNeasy Tissue kit (Qiagen 69504) with Gram-positive pre-treatment, according to the manufacturer's instructions. Paired-end multiplex libraries were created for each strain and sequenced using the Illumina MiSeq or NextSeq platform with Nextera XT libraries. Draft genome sequences were generated using SKESA version 2.2[37] or Spades v1.09[38] with default parameters. Gene predictions and annotations were generated using Prokka v 1.14 and streptococcal RefSeq-specific databases as described[27]. AMR genes were detected using Abricate v 0.8.2 (https://github.com/tseemann/abricate). Screens for *thfT*, *dfrF* and *dfrG* were performed against a diverse dataset of 2083 genomes for GAS and 136 publicly available genomes from refseq for *S. dysgalactiae*, using screen_assembly[27].

### Mutation detection
To determine the changes in serial passaged strains TB08 paired-end reads of both the parent strain and serial passaged strain from days 3, 14, and 15 were mapped to the TB08 assembled draft genome using Breseq version 0.35.1[39].

## Metabolic rescue assay

Metabolic rescue of GAS in the presence of antibiotics was performed using compounds listed in Supplementary Table 6. A 0.5 McFarland standard was prepared in 0.9% w/v NaCl following the overnight growth of bacterial strains on HBA. Unless otherwise mentioned, each compound was added to a 1 ml aliquot of each bacterial suspension to achieve a final concentration of 31.8 µg/ml, and 100 µL (equivalent to 50 ng/cm$^2$) was spread over the surface of 90 mm round Petri plates. An Etest was then performed to determine the MICs, as described above.

## Heterologous expression of *thfT* in GAS

Expression of *thfT* in heterologous GAS strains was performed using plasmid pJRS9508[40]. For expression under the P23 promoter, the vector was prepared by PCR using primers pJRS9508-fwd (5′-TCG AGA CTA AAG CAG AAG-3′) and pJRS9508 + P23-rev (5′-CAA CAT CAT TGT CAT TCA TAT TTT TC-3′), and *thfT* alleles were amplified using primers thfT+P23-fwd (5′-TAT GAA TGA CAA TGA TGT TGC AGT GAC AAA AAA ATG TTA CTC- 3′) and thfT-rev (5′-TAC TTC TGC TTT AGT CTC GAA AAA AGA AGA ACA AAG TGG G- 3′). For expression without the P23 promoter, the vector was amplified with primers pJRS9508-fwd and pJRS9508-P23-rev (5′-CTA TTT AAT CAC TTT GAC TAG GC-3′), and *thfT* alleles were amplified with primers thfT-P23-fwd (5′-TAG TCA AAG TGA TTA AAT AGC AGT GAC AAA AAA ATG TTA CTC-3′) and thfT-rev. PCR was performed using the KAPA HiFi PCR kit (Roche) and PCR products were purified using Monarch® PCR & DNA Cleanup Kit (New England Biolabs T1030S) or Monarch® DNA Gel Extraction Kit (New England Biolabs T1020S). Cloning was performed using the NEBuilder HiFi assembly kit (New England Biolabs E5520S) according to the manufacturer's instructions. Plasmids were purified using the Monarch Plasmid Miniprep kit (New England Biolabs T1010L), and sequences verified by whole-plasmid sequencing (PlasmidSeq, Genomics WA). Plasmids were transformed into electrocompetent GAS strains using the following settings: voltage = 1.75 kV, resistance 400 Ω, capacitance = 25 µF[41]. Transformants were selected by plating on THYA containing 100 µg/ml spectinomycin (THYA + Sp100).

Insertion of *thfT* into the genome of NS5347 was performed using the temperature-sensitive plasmid pLZts[41] (Addgene # 128799). The *thfT* gene and ~2.5 kb flanking DNA was amplified by PCR from TB08 DNA using primers pLZ-thfT-S (5′-CAT AAC CTG AAG GAA GAT CTA AAA CCA TAA TTT CCT TTC G-3′) and pLZ-thfT-A (5′-GTC GTC AGA CTG ATG GGC CCA GTA AAA CTC AAT GAG AAA ATA TTT TTT TAG-3′), and cloned into pLZts amplified by PCR with primers pLZts-S (5′-AGA TCT TCC TTC AGG TTA TG-3′) and pLZts-A (5′-GGG CCC ATC AGT CTG ACG-3′). Fragments were combined using the NEBuilder HiFi assembly kit as above. The resulting plasmid was transformed into electrocompetent NS5347 GAS cells as described above and selected by plating on THYA + Sp100 at 28 °C. Insertion of *thfT* into the genome of NS5347 was accomplished by homologous recombination, following plasmid integration at 37.5 °C and plasmid excision at 28 °C[41]. Colonies containing the desired allelic exchange were screened for the presence of *thfT* by PCR, using primers thfT+222 S (5′-GCC AGT GGG TCA GGT AAT TT-3′) and thfT+524 A (5′-CTT TGA AGG GCT GGC ACT AAT A-3′).

## Quantitative gene expression analysis

A single colony of each strain was cultured overnight in THYB at 37 °C, and resuspended in the same medium at an $OD_{600}$ of 0.01. Cells were harvested at mid-exponential ($OD_{600}$ = 0.4–0.6) and stationary ($OD_{600}$ = 1.2–1.4) phases and added to two volumes of RNAprotect (Qiagen 76506). Total RNA was then isolated using the RNeasy minikit (Qiagen 74104) and the samples were treated with Turbo DNase (Life Technologies AM1907) to remove any genomic DNA. Conversion of RNA to cDNA was performed using SuperScript VILO cDNA synthesis kit (Invitrogen 11754050). Reverse transcription-quantitative PCR (RT-qPCR) was performed using SYBR green master mix (Applied Biosystems 4309155) according to the manufacturer's instructions and primers specific for *thfT* (thfT-qPCR-S1: 5′-GCC AGT GGG TCA GGT AAT TT-3′; thfT-qPCR-A1: 5′-GAC AGT GGT TTC CGG TAG AAG-3′) and *gyrA* (RTPCR-gyrA-fw: 5′-CGA CTT GTC TGA ACG CCA AA-3′; RTPCR-gyrA-rv: 5′- GTC AGC AAT CAA GGC CAA CA-3′)[26]. qPCR was performed on a CFX96 Touch Real-Time PCR machine (BioRad) and analysed using CFX Manager software (BioRad). Relative gene expression was calculated using the threshold cycle ($2^{-\Delta\Delta Ct}$) method with *gyrA* as the reference housekeeping gene. All reactions were performed in triplicate from three independently isolated RNA samples for each strain.

## Statistical analysis

All statistical analysis was performed with Graphpad Prism (v9.3.1) or Microsoft Excel for Mac (v16.55) software unless otherwise indicated.

## Reporting summary

Further information on research design is available in the Nature Research Reporting Summary linked to this article.

## Data availability

The GAS whole genome sequencing data generated in this study have been deposited in the National Center for Biotechnology Information Sequence Read Archive database under accession code PRJNA791833. The metabolomic profiling data generated in this study have been deposited in the Mendeley Data database under accession code h4tkyd6vwj. GAS genes mapped onto the One carbon pool by folate reference pathway was performed at the KEGG Pathway Database (https://www.kegg.jp/pathway/map00670). The processed source data generated in this study are provided in the Source Data file. All other data supporting the findings of this study are available from the corresponding author on reasonable request. Requests for materials should be addressed to T.C.B. Source data are provided with this paper.

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

## Acknowledgements

We thank Manfred Rohde and Mark Walker for providing bacterial strains, Caitlyn Richworth and Jana Haasbroek for help with experiments, Steve McMaster for supplying reagents, Jean Lee and Ben Howden for providing the gradient agar plate method, Ritika Kar Bahal and Mark Nicol for help with photographing agar plates, and Julie Marsh for statistics advice. Some figure panels were created with Biorender.com. This work was supported by the National Health and Medical Research Council grants 1131932 (JRC), 1145033 (SYCT) and 1165876 (MRD), Western Australia Child Research Fund round 8 (TCB), Telethon Perth Children's Hospital Research Fund round 6 (TCB), Wesfarmers Centre of Vaccines and Infectious Diseases (TCB). TCB is supported by a fellowship from the Western Australian Future Health & Innovation Fund and was previously supported by the Australian National Health and Medical Research Council (NHMRC)-funded 'Improving Health Outcomes in the Tropical North: A multidisciplinary collaboration (HOT NORTH)', grant 1131932. MRD is supported by the University of Melbourne CR Roper Fellowship.

## Author contributions

P.M.G., A.C.B. and T.C.B. conceptualised the study. M.K.D.R., K.A.S., M.R.D. and T.C.B. developed methods. M.P.G.vdL. provided resources. M.K.D.R., A.S., A.J.H., A.M.W., J.A., J.L.P., J.I., J.H., S.W., T.W., P.N., J.A.L., K.J.B., N.G., M.R.D. and T.C.B. carried out the investigation. S.Y.C.T., M.R.D. and T.C.B. supervised the study. J.R.C., A.C.B., M.R.D. and T.C.B. acquired the funding. M.K.D.R., A.S., A.J.H., M.R.D. and T.C.B. wrote the original draft. All authors edited the manuscript and approved the final submission.

## Competing interests

The authors declare no competing interests.

## Additional information

[1]Wesfarmers Centre of Vaccines and Infectious Diseases, Telethon Kids Institute, University of Western Australia, Perth, WA, Australia. [2]Department of Microbiology and Immunology at the Peter Doherty Institute for Infection and Immunity, University of Melbourne, Melbourne, VIC, Australia. [3]Australian National Phenome Centre and the Centre for Computational and Systems Medicine, Health Futures Institute, Murdoch University, Perth, WA, Australia. [4]Department of Infectious Diseases at the Peter Doherty Institute for Infection and Immunity, University of Melbourne, Melbourne, VIC, Australia. [5]School of Molecular Sciences, University of Western Australia, Perth, WA, Australia. [6]German National Reference Center for Streptococci, Institute of Medical Microbiology, University Hospital RWTH Aachen, Aachen, Germany. [7]Global and Tropical Health Division, Menzies School of Health Research, Charles Darwin University, Darwin, NT, Australia. [8]College of Health and Human Sciences, Charles Darwin University, Darwin, NT, Australia. [9]Victorian Infectious Diseases Service, The Royal Melbourne Hospital, Melbourne, Australia. [10]Department of Infectious Diseases, Perth Children's Hospital, Perth, WA, Australia. [11]The Marshall Centre for Infectious Diseases Research and Training, School of Biomedical Sciences, University of Western Australia, Perth, Australia. ✉e-mail: timothy.barnett@telethonkids.org.au

