## [Peer Review File · Nature Communications]

Host-dependent resistance of Group A *Streptococcus* to sulfamethoxazole mediated by a horizontally-acquired reduced folate transporterREVIEWER COMMENTS

Reviewer #1 (Remarks to the Author):

Summary:

The paper by Rodrigo et al identifies a new mechanism by which resistance to sulfamethoxazole is acquired, specifically attributed to an energy-coupling factor (ECF) transporter S component gene, *thfT*. They show this gene enables Group A Streptococcus to acquire extracellular tetrahydrofolate and the end products of folate biosynthesis, which renders sulfamethoxazole resistance. They also show that THfT-mediated resistance is otherwise undetectable via routine surveillance methods, and thus is significant in understanding antibiotic susceptibility during infection.

Response:

The authors seem to have found a novel mechanism by which resistance to sulfamethoxazole occurs. The authors do a great job at contextualizing the phenomena of AMR, and describing the interplay between sulfamethoxazole (SMX), trimethoprim (TMP), and co-trimoxazole (SXT) resistance; also, their results highlight how diverse methodologies can be used to characterize antibiotic resistance, which is important. However, there are several limitations both in the methodology and justifications. In many cases, the simplest experimental validations were not provided, and instead the results appear convoluted and logically disconnected. Thus, while I think the paper is overall really strong, and the authors have found a really interesting phenomenon, these limitations very much dampen my overall enthusiasm and prevent me from feeling confident in the conclusions.

Major points:

- 1) Overall, the text needs to be improved for readership for the many jumps in logic throughout. The proposed mechanism is extremely complicated to follow for the broad audience of this journal, and therefore the rationale/expectations behind each experiment is not always clear. The authors should provide the diagram presented in Fig. 2 earlier in the text, and explain as they go by calling out specific arrows as they relate to their predictions (for example, where does folic acid feed into this cycle?). A fresh eye will likely go a long way in addressing many of these gaps in logical reasoning.
- 2) More generally, several results are difficult to interpret due to the experimental designs used. The susceptibility testing methods keep changing; considering that is the primary quantification method, these need to be far better justified and cross-validated in many cases. There are also missing details on replicates and stats. Overall, confidence in the conclusions would significantly increase if the methods were more rigorously justified.
- 3) In line with points 1-2, although many aspects can likely be clarified by updating the text, there are several instances where jumps in logic leave missing controls. For example, and perhaps most fundamental, is in lines 116-119 where the authors speculate that the 5 bp duplication in the promoter resulted in increased expression of *thfT*, but never verify directly in their evolved strain (TB08-2-14) – this should be easy and straight forward to do before jumping to more complicated implications. Also, they should verify changes in export/import rates or intracellular concentrations of THF (or reduced extracellular).
- 4) Lines 66-70: The authors attribute differences between Etest and microdilution results between differences in growth media. Yet, these tests fundamentally differ in many other ways, including solid agar vs liquid media (which could have diffusion implications), as well as the continuous vs discrete gradient of the drug. The authors should update the text to reflect this (we don't yet know until later that nutrient uptake is a main focus). Moreover, it's not even clear to me that this is the reason behind the large discrepancy in resistance numbers even after reading the full paper.
- 5) Lines 120-131: The results presented in this paragraph were perhaps the most difficult to get behind. Can the authors verify THF presence/absence in the media? Why did intermediate levels of THF provide the greatest resistance? One concentration/replicate with one type of MIC is not sufficient to demonstrate this, especially considering the variability they found between tests. The authors should include an actual dose response to demonstrate the evolved strains' increased resistance at lower THF concentrations, and they should do this using more than 1 MIC method. Also, the authors should explain the intuition behind folic acid not providing any resistance- at face value this latter point seems counterintuitive.
- 6) Lines 134-135: Doesn't the reliance on *dfrG* and *dfrF* for *thf*-mediated resistance suggest a more complex mechanism than just THF transport alone? Can the authors elaborate on this?

Reviewer #2 (Remarks to the Author):

Rodrigo et al present the discovery of a new antimicrobial resistance mechanisms that depends on the presence of an exogenous (host) metabolite (THF), and a membrane transporter that can import the compound (an ECF transporter). I find the work very interesting but I must emphasize that my expertise is not in the AMR field. From the perspective of my own expertise in membrane transport, I find the work also intriguing, but suggest to make the presentation more accurate.

Major points.

1. I find the number of replicates used in the experiments and the statistic treatment of the results somewhat meagre. In many cases a single "representative" experiment is shown out of two or three replicates (the data of which are not shown). In some cases only a single experiment was done (extended data table 2) I do not know what is common practice in the AMR field, but in my field this would not be acceptable. I believe that it would be straightforward to amend.

2. Not being from the AMR field, I find the manuscript not easy to read. It has be written up in a very condensed manner, with very little explanation.

- I had to read lines 58-70 many times before I could understand. For example Etest is a brand name; the difference between MHF and broth medium is not explained; In extended Table 1 it is unclear what MLST and emmST are; looking at the table, I see 24 strains (not 22, although I can infer that the upper two strains are control strain, but this is not explained); I see 7 red shaded cells in the STX MIC column (not 6), but only later the authors explain that the cutoff for resistance is 2 ug/ml. This is confusing. Also in this table, there is no information on replicates and statistics.

- Extended data fig 2 is impossible to understand without explanation. Which mutations are relevant?

- I suggest to make the text more accessible to readers from outside the AMR field, with a proper introduction section, figure legends that contain all the information etc.

3. The description of the ECF transporters is not accurate, and references are not chosen well.

- I find the crosstalk between a horizontally transferred S-component and an endogenous ECF module intriguing. Such crosstalk has been eluded to (DOI: 10.1128/JB.01208-08) This paper must be cited, because it forms the basis of what we know about ECF transporters.

- The references to crystal structures (ref 14 and 15) are not suitable to introduce the peculiarity of ECF transporters. Crystal structures are really not relevant here. Much more relevant are (DOI: 10.1128/JB.01208-08), where the basis of interaction of multiple S-components with a single ECF module is presented, and doi: 10.7554/eLife.64389 where it was experimentally shown. I also suggest to cite a recent review.

- In the introduction the authors state that "ThfT likely modifies the substrate specificity of an ECF transporter". Perhaps just semantics, but it would be more accurate to state that it expands the substrate spectrum. S-components are the sole determinants of substrate specificity in ECF transporters, yet not sufficient for transport. Streptococcus encodes many different S-components, each with a different substrate spectrum. Each S-component interacts with the same (endogenous) ATP-coupling complex (the ECF module). An extra S-component, horizontally transferred will therefore expand the substrate spectrum.

Minor:

Line 193 the word "prototype" seems weird and suggest an engineering effort

Fig 2 panel B: The figure suggests that the pdb structure of an ECF transporter was used to make the cartoon of the blue/grey/purple protein in the membrane, but it looks nothing like an ECF transporter! Especially, S-components are integral membrane proteins. Biorender can make accurate icons directly from pdb codes.

Reviewer #3 (Remarks to the Author):

In this study, the authors discovered that a transporter component (thfT) enables Group A streptococcus to acquire extracellular tetrahydrofolate (THF) and confers resistance to sulfamethoxazole in the presence of exogenous THF.

Their findings are novel and clinically relevant. I have several suggestions for the authors to consider.

1. The authors should acknowledge that sulfamethoxazole is not the major antibiotic used to treat group A streptococcus infections. Also, group A streptococcus as a species is, in general, not highly susceptible to sulfamethoxazole. This is why sulfamethoxazole is supplemented to the group A streptococcus selective agar (Gunn et al., J Clin Microbiol. 1977).
2. The authors should examine group A streptococcus (thfT positive and negative strains) sensitivity to sulfamethoxazole in the host-relevant environment, such as whole blood, serum, saliva, or animal infection models. This dataset is essential to support the conclusion that this transporter mediates "host-dependent resistance".
3. The authors should clarify in their text that the majority of the group A streptococcus clinical isolates do not have the horizontally acquired thfT gene.

Response to reviewer's comments for NCOMMS-22-09046-T (Rodrigo *et al.*)

REVIEWER COMMENTS

Reviewer #1 (Remarks to the Author):

Summary:

The paper by Rodrigo *et al* identifies a new mechanism by which resistance to sulfamethoxazole is acquired, specifically attributed to an energy-coupling factor (ECF) transporter S component gene, *thfT*. They show this gene enables Group A Streptococcus to acquire extracellular tetrahydrofolate and the end products of folate biosynthesis, which renders sulfamethoxazole resistance. They also show that THfT-mediated resistance is otherwise undetectable via routine surveillance methods, and thus is significant in understanding antibiotic susceptibility during infection.

Response:

The authors seem to have found a novel mechanism by which resistance to sulfamethoxazole occurs. The authors do a great job at contextualizing the phenomena of AMR, and describing the interplay between sulfamethoxazole (SMX), trimethoprim (TMP), and co-trimoxazole (SXT) resistance; also, their results highlight how diverse methodologies can be used to characterize antibiotic resistance, which is important. However, there are several limitations both in the methodology and justifications. In many cases, the simplest experimental validations were not provided, and instead the results appear convoluted and logically disconnected. Thus, while I think the paper is overall really strong, and the authors have found a really interesting phenomenon, these limitations very much dampen my overall enthusiasm and prevent me from feeling confident in the conclusions.

We thank the reviewer for their positive comments. We have addressed each of the reviewer's specific comments below.

Major points:

1) Overall, the text needs to be improved for readership for the many jumps in logic throughout. The proposed mechanism is extremely complicated to follow for the broad audience of this journal, and therefore the rationale/expectations behind each experiment is not always clear. The authors should provide the diagram presented in Fig. 2 earlier in the text, and explain as they go by calling out specific arrows as they relate to their predictions (for example, where does folic acid feed into this cycle?). A fresh eye will likely go a long way in addressing many of these gaps in logical reasoning.

We have undertaken a major rewrite to improve the readability for the broad audience of *Nature Communications*. Specifically, we have reformatted to manuscript from a letter format to an article format and included additional text and schematic figures to explain certain concepts and link the logical flow of ideas. As suggested by the reviewer, we have also inserted a new figure (new Fig. 1) outlining folate metabolism in bacteria and mammals and how these differ, and highlighted where folic acid feeds into this cycle.

2) More generally, several results are difficult to interpret due to the experimental designs used. The susceptibility testing methods keep changing; considering that is the primary quantification method, these need to be far better justified and cross-validated in many cases.

We agree. During our characterization of ThfT-mediated resistance we experienced multiple extensive delays in the shipment of reagents, including BioMerieux ceasing shipment of culture media to Australia in mid-2021. We have since negotiated with a local distributor to obtain BioMerieux media directly from Europe and have repeated all Etest results to standardise MIC values across this study. To clarify results that are media-dependent, we have repeated selected MIC experiments by Etest with MHF media from both BioMerieux and Oxoid. Where required to resolve MICs greater than the range measurable by Etest, we have performed additional MIC assays and provided justification for these in the text.

There are also missing details on replicates and stats. Overall, confidence in the conclusions would significantly increase if the methods were more rigorously justified.

We have added information in the text on experimental replicates and provided statistical analyses where appropriate. We have also provided all data in the Supplementary Data file.

3) In line with points 1-2, although many aspects can likely be clarified by updating the text, there are several instances where jumps in logic leave missing controls. For example, and perhaps most fundamental, is in lines 116-119 where the authors speculate that the 5 bp duplication in the promoter resulted in increased expression of *thfT*, but never verify directly in their evolved strain (TB08-2-14) – this should be easy and straight forward to do before jumping to more complicated implications.

We thank the reviewer for this suggestion. We have performed this additional experiment (new Fig. 2D) and described this data in the text as follows:

Lines 176-184: "The 5 bp duplication in TB08-2-14 is in the intergenic region between *thfT* and the divergently transcribed gene *ddlA*. As this intergenic region likely contains the promoter region for *thfT*, we next examined whether the increased SXT resistance in TB08-2-14 was associated with upregulation of the *thfT* gene. RNA was prepared from each strain and *thfT* mRNA levels were quantified by qRT-PCR (Fig. 2D). Results from this experiment demonstrate that the 5 bp duplication in TB08-2-14 is associated with an increased level of *thfT* transcript in both mid-exponential (16.6-fold) and stationary (3.8-fold) phases of growth relative to TB08, and suggests that *thfT* could be a novel AMR gene that is up-regulated in TB08-2-14 due to a mutation in the *thfT* promoter region."

Also, they should verify changes in export/import rates or intracellular concentrations of THF (or reduced extracellular).

In an effort to address this comment and point 5 (below), we first attempted to quantify reduced folates in the growth medium from each supplier (MH-Bm and MH-Ox) by NMR and LC-MS/MS (outlined in Supplementary Note 1 and associated Figures and Table). While we were able to identify differences in several components of the media from each supplier, we were unable to detect substantial differences in the concentration of reduced folates that

would explain the relative performance of these media. We believe this is due to the sensitivity of detection of reduced folates (limit of detection >100 ng/ml) with the high-resolution mass spectrometer instrument that we used relative to the concentration of THF required to rescue ThfT-dependent growth in the presence of SMX (~20 ng/ml). The only other method we could potentially use to measure import is using radiolabelled compounds, however the only commercially available folate compound we are able to source is ³H-folic acid, which does not function in our system. For these reasons, we did not attempt further quantification of reduced folates in media or bacterial cells post bacterial growth.

We believe the reviewer's comment relates to whether we have sufficient experimental evidence that ThfT is functioning as a reduced folate transporter. Further, there appears to have been some slight confusion with our data on the relative amounts of extracellular THF required to rescue growth of the wild type (TB08) and evolved (TB08-2-14) strains in the presence of co-trimoxazole (Reviewer 1 point 5, below), which we have now addressed. With the combined evidence presented in the manuscript, including the additional experiments suggested by the reviewer, we believe that we have provided sufficient evidence to support that ThfT is acting as a reduced folate importer as follows: 1) Homology of ThfT with FolT, which has been experimentally shown to specifically import radiolabelled folate (Reference 20, Rodionov et al.); 2) the lower amount of THF required to rescue growth when ThfT is upregulated in the evolved strain TB08-2-14 (EC₅₀ 3.37 ng/cm²) compared with the parent strain TB08 (EC₅₀ 24.7 ng/cm²; Fig. 3A); and 3) The ability of multiple extracellular reduced folates to rescue growth in a ThfT-mediated manner (Fig. 5A).

4) Lines 66-70: The authors attribute differences between Etest and microdilution results between differences in growth media. Yet, these tests fundamentally differ in many other ways, including solid agar vs liquid media (which could have diffusion implications), as well as the continuous vs discrete gradient of the drug. The authors should update the text to reflect this (we don't yet know until later that nutrient uptake is a main focus). Moreover, it's not even clear to me that this is the reason behind the large discrepancy in resistance numbers even after reading the full paper.

We agree. We have updated the text to reflect that the differences in MICs observed in our study with published values could reflect a number of important differences between Etest and microdilution methodologies, as follows:

Lines 97-99: "This discrepancy is potentially explained by methodological differences between these assays, or differences in the composition of the growth medium used in each assay."

5) Lines 120-131: The results presented in this paragraph were perhaps the most difficult to get behind. Can the authors verify THF presence/absence in the media?

See response to point 3, above.

Why did intermediate levels of THF provide the greatest resistance? One concentration/replicate with one type of MIC is not sufficient to demonstrate this, especially considering the variability they found between tests. The authors should include an actual

dose response to demonstrate the evolved strains' increased resistance at lower THF concentrations, and they should do this using more than 1 MIC method.

We agree with the reviewer that the results presented in this paragraph are confusing. Our intention was to describe that an intermediate level of THF enabled us to resolve a difference in SXT resistance between the parent (TB08) and evolved (TB08-2-14) strains due to more efficient transport of THF by the evolved strain. As suggested by the reviewer, we have performed a dose-response experiment to demonstrate that the evolved strain has increased resistance at lower THF concentrations by Etest (Fig. 3A) and broth microdilution (Supplementary Fig. 6), and provided expanded explanation in the text as follows:

Lines 195-213: " While the *thfT*-negative strain M6^{JRS4} did not exhibit any change in SXT MIC in the presence of either compound, both TB08 and TB08-2-14 showed an increased SXT MIC in the presence of extracellular THF but not folic acid, with both strains being resistant to >32 µg/ml SXT at THF concentrations >50 ng/cm² THF. The lack of resistance in the presence of extracellular folic acid likely reflects that folic acid is a very poor substrate for most bacterial dihydrofolate reductase enzymes²³. A dose-response comparison by Etest showed that the amount of THF required to increase SXT resistance in the evolved strain TB08-2-14 (EC₅₀ = 3.37 ng/cm²; Fig. 3A) was lower than the amount required for the parent strain TB08 (EC₅₀ = 24.7 ng/cm²; Fig. 3A). Similar results were obtained by broth microdilution assay (TB08-2-14 EC₅₀ = 1.72 ng/ml, TB08 EC₅₀ = 7.60 ng/ml; Supplementary Fig. 6). These results are likely explained by the increased levels of ThfT in TB08-2-14, which may then outcompete other ECF S components for the same ECF transport module (Fig. 3C). This is supported by recent evidence that ECF S components can compete for the same ECF transport module, with high levels of substrate for one S component (e.g. pantothenate) able to reduce uptake of second substrate (e.g. folate) that is transported by an alternative S component²⁴. Together, these data indicate that THF or related compounds are likely present in MH agar sourced from Oxoid at an intermediate concentration sufficient to support SXT resistance for TB08-2-14 but not for TB08 (Fig. 3A). In contrast, these compound(s) are absent or at much lower concentrations in MHF agar from BioMerieux (MHF-Bm) and are unable to support SXT resistance for either strain."

Also, the authors should explain the intuition behind folic acid not providing any resistance- at face value this latter point seems counterintuitive.

Folic acid is not a substrate for bacterial dihydrofolate reductase enzymes, and thus is unable to be reduced to DHF. We have highlighted this in new Fig. 1, and provided an explanation in the figure legend and the text, as follows:

Lines 198-200: " The lack of resistance in the presence of extracellular folic acid likely reflects that folic acid is a very poor substrate for most bacterial dihydrofolate reductase enzymes²³."

6) Lines 134-135: Doesn't the reliance on *dfrG* and *dfrF* for *thf*-mediated resistance suggest a more complex mechanism than just THF transport alone? Can the authors elaborate on this?

We agree. SXT resistance requires both SMX and TMP resistance mechanisms. Our results show that ThfT is a SMX-specific resistance mechanism, that alone provides high-level SMX

resistance but requires an independent TMP resistance mechanism (*dfrG/F*) for SXT resistance. We have clarified this in the text as follows:

Lines 216-223: "Comparative genomics of 24 globally sourced GAS strains (Supplementary Table 1) revealed five strains that encoded *thfT*. To investigate whether all strains that encoded *thfT* were SXT resistant with exogenous THF, we next measured SXT MICs of each strain in the absence and presence of 50 ng/cm² THF (Supplementary Table 2). Results showed that each of the *thfT*-positive strains that also encoded a TMP resistance gene (*dfrG* or *dfrF*) and were TMP resistant (>32 µg/ml) became highly-resistant to SXT in the presence of exogenous THF, while both of the strains that lacked a TMP resistance gene exhibited a modest (~4 to 5-fold) increase in SXT MIC values in the presence of THF."

To further describe the role of ThfT if conferring SMX resistance, we expressed *thfT* in a GAS strain that lacks SMX or TMP resistance and measured MICs for SMX, TMP and SXT, and clarified in the text as follows:

Lines 235-249: " To further investigate whether ThfT conferred SXT resistance or SMX resistance, we expressed each ThfT variant in a laboratory GAS strain (M6^{JRS4}) that lacked any TMP- or SMX-resistance mechanism, and measured the effect of exogenous THF on MIC values for SMX, TMP and SXT (Fig. 4A, Supplementary Table 4). Expression of either variant resulted in a very modest increase in SXT MIC in the presence of 50 ng/cm² of exogenous THF, yet had no effect on MIC values obtained for TMP. However, expression of either variant resulted in high level SMX resistance (>1024 µg/ml) even in the absence of additional extracellular THF. This result was unexpected since addition of exogenous THF is required for SXT resistance in the parent strain (Fig. 3A,B). We rationalised that the lack of a requirement for additional THF for SMX resistance in this experiment may be explained by higher expression levels of *thfT* when expressed on a multicopy plasmid, which may then require lower levels of exogenous THF to bypass SMX inhibition of the folate synthesis pathway. The requirement of only small amounts of extracellular THF is likely explained by the fact that THF is recycled back to DHF during the production of thymidine monophosphate, such that only very low levels of THF should theoretically support DNA synthesis (Fig. 4B)."

Finally, to control for any gene dosage effects on SMX resistance caused by expression on a multicopy plasmid, we inserted *thfT* into the chromosome of a strain (NS5347) that lacks *thfT* and is sensitive to both SMX and TMP, and described this experiment in the text as follows:

Lines 250-261: " To further investigate the requirement of extracellular THF for ThfT-mediated SMX resistance and control for any experimental artefacts caused by gene dosage and strain background, we next inserted the *thfT* T71 variant into the genome of a *thfT*-negative GAS strain by homologous recombination. For this experiment, we used NS5347 as the parent strain as it belonged to the same multi-locus sequence type (MLST 289) as TB08 and had the same nucleotide sequences flanking the *thfT* locus. This enabled efficient recombination of *thfT* into exactly the same locus as in TB08. Examination of each strain in the absence and presence of 50 ng/cm² exogenous THF revealed that *thfT* was able to confer high levels of SMX resistance (>1024 µg/ml), but only in the presence of exogenous THF (Fig. 4C, Supplementary Table 5). This resulted in a very modest increase in SXT MIC (~4.5-fold) with no associated increase in TMP MIC. These results demonstrate that ThfT functions as a SMX

resistance determinant that imports extracellular THF and bypasses the inhibitory activity of SMX on the folate biosynthesis pathway (Fig. 4B)."

Reviewer #2 (Remarks to the Author):

Rodrigo et al present the discovery of a new antimicrobial resistance mechanisms that depends on the presence of an exogenous (host) metabolite (THF), and a membrane transporter that can import the compound (an ECF transporter). I find the work very interesting but I must emphasize that my expertise is not in the AMR field. From the perspective of my own expertise in membrane transport, I find the work also intriguing, but suggest to make the presentation more accurate.

We thank the reviewer for their positive comments.

Major points.

1. I find the number of replicates used in the experiments and the statistic treatment of the results somewhat meagre. In many cases a single “representative” experiment is shown out of two or three replicates (the data of which are not shown). In some cases only a single experiment was done (extended data table 2) I do not know what is common practice in the AMR field, but in my field this would not be acceptable. I believe that it would be straightforward to amend.

As suggested, we have performed additional experimental replicates to include at least three biological replicates for each dataset. It is not uncommon in the AMR literature for single MIC replicates to be reported when comparing of strains (e.g. Lee et al. DOI: 10.1038/s41564-018-0230-7). However, we appreciate the reviewer's concern and have conducted additional experimental replicates to include at least three biological replicates for each dataset. To improve clarity and readability, we have reported median MIC values in tables and provided reference to the supplementary tables with statistical analyses, and provided a supplementary dataset that includes all replicates.

For many of the datasets in our manuscript the MIC values are greater than the resolution of the assay (e.g. with results presented as >32 ug/ml for SXT). As the resistant phenotype has no variance in MIC values, we have used a one sample t test to compare groups where one or more groups have no variance due to the resolution of the assay. This was based on advice from an experienced biostatistician (Dr Julie Marsh, Telethon Kids Institute). Details of these analyses are provided in the appropriate figure legend and the Methods.

2. Not being from the AMR field, I find the manuscript not easy to read. It has be written up in a very condensed manner, with very little explanation.

As suggested by the reviewer, we have undertaken a major rewrite to improve the readability for the broad audience of *Nature Communications*. Specifically, we have reformatted to manuscript from a letter format to an article format, and used the extra space to include additional text to explain certain concepts and link the logical flow of ideas.

- I had to read lines 58-70 many times before I could understand. For example Etest is a brand name; the difference between MHF and broth medium is not explained; In extended Table 1 it is unclear what MLST and emmST are; looking at the table, I see 24 strains (not 22, although I can infer that the upper two strains are control strain, but this is not explained); I see 7 red

shaded cells in the STX MIC column (not 6), but only later the authors explain that the cutoff for resistance is 2 ug/ml. This is confusing. Also in this table, there is no information on replicates and statistics.

We thank the reviewer for their comments and have made substantial changes to the organisation of the manuscript, including the following changes to clarify this point:

Lines 74-80: "All described resistance mechanisms to TMP and SMX involve mutations in the target enzymes of each antibiotic (dihydrofolate reductase and dihydropteroate synthase, respectively), or horizontally-acquired enzyme variants with reduced antibiotic affinity¹¹. While genetic mechanisms of TMP and SMX resistance have been described in clinical GAS isolates^{6,12-14}, the combinations that lead to SXT resistance remain unclear. In this study we investigated SXT resistance in clinical GAS isolates and identified a new mechanism of SMX resistance that involves acquisition of reduced folates from the extracellular environment."

Lines 90-99: "***In vitro* resistance of GAS to SXT requires high level TMP resistance conferred by *dfrF*.** To investigate the combinations of TMP and SMX resistance that confer SXT resistance, we determined the susceptibility for SMX, TMP and SXT for a global collection of GAS strains reported to have reduced susceptibility to SXT^{13,15} (Supplementary Table 1). The minimal inhibitory concentration (MIC) for each strain was determined using Epsilometer test (Etest) strips and Mueller-Hinton Fastidious agar (MHF; BioMerieux). While each strain was originally reported as being SXT-resistant using broth microdilution methodology^{13,15}, only 6/22 were SXT resistant using Etest methodology. This discrepancy is potentially explained by methodological differences between these assays, or differences in the composition of the growth medium used in each assay."

We have also provided information in the legend of Supplementary Table 1 to define the terms MLST and *emmST*, improve the use of colour for MIC values, and provided information on replicates. As this table is a comparison of clinical strains, we have not undertaken statistical analyses as is the convention in the field (e.g. DOI: 10.1038/s41564-018-0230-7).

- Extended data fig 2 is impossible to understand without explanation. Which mutations are relevant?

We agree. We have amended the figure (now Supplementary Fig. 1) to highlight mutations important for SMX resistance. We have also included additional text to clarify the interpretation of FoLP sequences and SMX resistance, as follows:

Lines 105-118: "While several strains were found to have SMX MICs >100 µg/ml that were sufficient for SXT resistance, we were unable to identify any horizontally-acquired dihydropteroate synthase variants that would account for SMX resistance in these strains. Inspection of the sequence of the SMX target enzyme (FoLP) in each strain revealed considerable sequence variation, yet there were no individual amino acid sequence variations that correlated with reduced SMX susceptibility (Supplementary Fig. 1). Structural¹⁶ and biochemical¹⁷ investigations have identified key residues in dihydropteroate synthase enzymes involved in SMX drug binding and resistance, which correspond to residues F25, S26, T59 and P61 in the GAS FoLP sequence (Supplementary Fig. 1). Of these, only T59 showed any

variation between the strains investigated, however the two amino acid variations at this residue did not correlate with the observed SMX MICs for strains that contained them (T59A, MIC range 56 to >1024 ug/ml; T59N, MIC range 96 to >1024 ug/ml). Thus, while we can not discount that combinations of FolP amino acid changes might have contributed to a reduced SMX susceptibility in some of these strains, the exact mechanism of resistance remained unclear."

- I suggest to make the text more accessible to readers from outside the AMR field, with a proper introduction section, figure legends that contain all the information etc.

As suggested, we have undertaken a major rewrite of this manuscript to address this comment and similar comments from Reviewer 1 (point 1).

3. The description of the ECF transporters is not accurate, and references are not chosen well.

- I find the crosstalk between a horizontally transferred S-component and an endogenous ECF module intriguing. Such crosstalk has been elucidated (DOI: 10.1128/JB.01208-08) This paper must be cited, because it forms the basis of what we know about ECF transporters.

- The references to crystal structures (ref 14 and 15) are not suitable to introduce the peculiarity of ECF transporters. Crystal structures are really not relevant here. Much more relevant are (DOI: 10.1128/JB.01208-08), where the basis of interaction of multiple S-components with a single ECF module is presented, and doi: 10.7554/eLife.64389 where it was experimentally shown. I also suggest to cite a recent review.

We thank the reviewer for these excellent suggestions, and have included the suggested citations and further explanation of the crosstalk, including new Fig. 3C, as follows:

Lines 169-172: "*thfT* encodes a protein with homology to FolT, the folate-specific substrate binding component (S component) of an energy-coupling factor (ECF) transporter^{20,21} from *Lactocaseibacillus casei*²² (28% identity, 46% similarity) and *Leuconostoc mesenteroides*²⁰ (23% identity, 45% similarity; Supplementary Fig. 5)."

Lines 200-209: "A dose-response comparison by Etest showed that the amount of THF required to increase SXT resistance in the evolved strain TB08-2-14 ($EC_{50} = 3.37 \text{ ng/cm}^2$; Fig. 3A) was lower than the amount required for the parent strain TB08 ($EC_{50} = 24.7 \text{ ng/cm}^2$; Fig. 3A). Similar results were obtained by broth microdilution assay (TB08-2-14 $EC_{50} = 1.72 \text{ ng/ml}$, TB08 $EC_{50} = 7.60 \text{ ng/ml}$; Supplementary Fig. 6). These results are likely explained by the increased levels of ThfT in TB08-2-14, which may then outcompete other ECF S components for the same ECF transport module (Fig. 3C). This is supported by recent evidence that ECF S components can compete for the same ECF transport module, with high levels of substrate for one S component (e.g. pantothenate) able to reduce uptake of second substrate (e.g. folate) that is transported by an alternative S component²⁴. "

- In the introduction the authors state that "ThfT likely modifies the substrate specificity of an ECF transporter". Perhaps just semantics, but it would be more accurate to state that it expands the substrate spectrum. S-components are the sole determinants of substrate specificity in ECF transporters, yet not sufficient for transport. Streptococcus encodes many different S-components, each with a different substrate spectrum. Each S-component

interacts with the same (endogenous) ATP-coupling complex (the ECF module). An extra S-component, horizontally transferred will therefore expand the substrate spectrum.

We agree with the reviewer's suggestion and have modified the text to incorporate this suggestion, as follows:

Lines 37-39: "ThfT likely expands the substrate specificity of an endogenous ECF transporter to acquire reduced folate compounds directly from the host, thereby bypassing the inhibition of folate biosynthesis by sulfamethoxazole. "

Lines 80-83: "This mechanism of resistance is mediated by a horizontally acquired energy-coupling factor (ECF) transporter S component (ThfT) that expands the substrate profile of an endogenous ECF transporter to include reduced folate cycle intermediates."

Lines 189-191: "Given the homology of ThfT to the substrate binding S component of an ECF folate transporter, we hypothesised that ThfT might expand the substrate profile of an endogenous ECF transporter to include end products of the folate biosynthesis pathway. "

Lines 329-331: "ThfT works by expanding the substrate profile of an endogenous ECF transporter to include the end products of the folate synthesis pathway. "

Minor:

Line 193 the word "prototype" seems weird and suggest an engineering effort

We agree and have modified the text as follows:

Lines 337-340: "We propose the ThfT-mediated SMX resistance is a new class of host-dependent AMR (Supplementary Fig. 10), and our findings provide the opportunity to monitor any emergence of *thfT*-positive GAS and treat these infections with alternative antibiotics."

Fig 2 panel B: The figure suggests that the pdb structure of an ECF transporter was used to make the cartoon of the blue/grey/purple protein in the membrane, but it looks nothing like an ECF transporter! Especially, S-components are integral membrane proteins. Biorender can make accurate icons directly from pdb codes.

We thank the reviewer for this suggestion. We have updated the figures using more accurate cartoons for each ECF transporter complex component, based on the available structural information for the Folt-ECF transporter complex (PDB: 7NNU) and Folt S-component (PDB: 5DOY).

Reviewer #3 (Remarks to the Author):

In this study, the authors discovered that a transporter component (thfT) enables Group A streptococcus to acquire extracellular tetrahydrofolate (THF) and confers resistance to sulfamethoxazole in the presence of exogenous THF.

Their findings are novel and clinically relevant.

We thank the reviewer for their positive comments.

I have several suggestions for the authors to consider.

1. The authors should acknowledge that sulfamethoxazole is not the major antibiotic used to treat group A streptococcus infections. Also, group A streptococcus as a species is, in general, not highly susceptible to sulfamethoxazole. This is why sulfamethoxazole is supplemented to the group A streptococcus selective agar (Gunn et al., J Clin Microbiol. 1977).

While sulfamethoxazole alone does not effective for treating GAS infections, it is effective in combination with trimethoprim (i.e. co-trimoxazole). This was shown both *in vitro* (Bowen et al JCM 2012 DOI: 10.1128/JCM.02195-12) and *in vivo* (Bowen et al Lancet 2014 DOI: 10.1016/S0140-6736(14)60841-2). Based on the results of this clinical trial, co-trimoxazole is now used across Australia and in the Pacific for treatment of impetigo. A systematic review from 2021 (Gahlawat et al. Clinical Therapeutics. DOI: 10.1016/j.clinthera.2021.04.013) also supports use of cotrimoxazole for impetigo.

To clarify the clinical use of co-trimoxazole for treatment of GAS infections, we have modified the text as follows:

Lines 71-73: "After promising clinical trial results⁸, SXT is now recommended to treat *Streptococcus pyogenes* (Group A *Streptococcus*, GAS) skin infections in endemic settings^{9,10}."

Lines 360-363: "While the combinations of TMP and SMX resistance genes required for SXT resistance are currently rare in global GAS strains, the apparent transmissibility of these genes means that emergence of SXT resistant GAS should be monitored with increasing use of this antibiotic to treat GAS skin infections."

2. The authors should examine group A streptococcus (thfT positive and negative strains) sensitivity to sulfamethoxazole in the host-relevant environment, such as whole blood, serum, saliva, or animal infection models. This dataset is essential to support the conclusion that this transporter mediates "host-dependent resistance".

We thank the reviewer for this excellent suggestion. To address this point, we have tested *thfT* isogenic strains for SMX resistance in the presence of epithelial cell lysates to mimic the environment of pus in impetigo lesions. Results of this experiment are presented in Fig. 5C,D, and explained in the text as follows:

Lines 295-305: "The above results show that ThfT enables uptake of multiple reduced folate compounds that comprise the intracellular folate pool in host cells. We next tested the ability of ThfT to acquire reduced folates directly from primary human epithelial cells. For this

experiment, we used a modified broth microdilution assay to allow measurement of SMX MICs in the presence of different concentrations of epithelial cell lysate (Fig. 5C). Results of this experiment show that the cellular folate pool is sufficient to rescue growth of NS5347::*thfT* but not the NS5347 parental strain (Fig. 5D). Thus, ThfT is able to acquire reduced folates from host cells and bypass SMX inhibition of folate synthesis. We propose that cellular folate cycle intermediates and one-carbon THF pools serve as multiple *in vivo* substrates for ThfT-mediated SMX resistance (Fig. 5B), which are likely released during infection from neutrophils recruited to the site of infection and/or tissue damage that occurs due to production of one or more GAS cytolytins^{25,26}."

3. The authors should clarify in their text that the majority of the group A streptococcus clinical isolates do not have the horizontally acquired *thfT* gene.

We agree. We have addressed this point as follows:

Lines 309-316: " To obtain a more comprehensive view on *thfT* carriage and SXT resistance in GAS strains, we investigated the presence of *thfT*, *dfrF* and *dfrG* in a global dataset of 2083 diverse GAS isolates²⁷ (Supplementary Table 8). Carriage of *thfT* was uncommon in this dataset (20/2083 GAS strains) yet maintained high levels of sequence homology. Furthermore, *thfT*-positive strains included 11 unique *emm* sequence types that were from multiple geographic regions. Of these, 6/20 *thfT*-positive strains also encoded a TMP resistance gene (*dfrF* or *dfrG*; Supplementary Table 8) and are likely functionally SXT resistant in the presence of extracellular reduced folate compounds."

Lines 360-367: " While the combinations of TMP and SMX resistance genes required for SXT resistance are currently rare in global GAS strains, the apparent transmissibility of these genes means that emergence of SXT resistant GAS should be monitored with increasing use of this antibiotic to treat GAS skin infections. Both *dfrG*¹² and *dfrF*³¹ have been found on genetic elements that are likely transferrable between GAS strains and related *Streptococcus* species, with *dfrF* located on integrative-conjugative elements along with multiple additional AMR genes. This raises the concern that *dfrF* could not only be selected for by increased use of SXT, but also with additional antibiotics."

REVIEWERS' COMMENTS

Reviewer #1 (Remarks to the Author):

The authors have done a fantastic job in clarifying my questions and conducting experiments to address my comments. The paper is very strong and will be a great addition to the field

Reviewer #2 (Remarks to the Author):

The manuscript has greatly improved. I have a few minor comments left:

1. line 96: 6/22 strains are STX resistant. In supplementary table 1 I really only see five red shaded cells (strains BB01, 09, 19, 31 and 34). Am I looking in the wrong place? Or is the number only 5?
2. line 216-217: please indicate the thfT strains in supplementary table 1
3. line 76: It would be good to introduce dhfG and dhfF here already (now they come a bit out of the blue later on) and possibly also acquired genes (other than thfT) that confer SMX resistance (if known)

Response to reviewer's comments for NCOMMS-22-09046-A (Rodrigo *et al.*)

REVIEWER COMMENTS

Reviewer #1 (Remarks to the Author):

The authors have done a fantastic job in clarifying my questions and conducting experiments to address my comments. The paper is very strong and will be a great addition to the field

We thank the reviewer for their enthusiastic comments and endorsement of our manuscript.

Reviewer #2 (Remarks to the Author):

The manuscript has greatly improved. I have a few minor comments left:

1. line 96: 6/22 strains are STX resistant. In supplementary table 1 I really only see five red shaded cells (strains BB01, 09, 19, 31 and 34). Am I looking in the wrong place? Or is the number only 5?

We thank the reviewer for highlighting this error and we have amended this in the text (Line 106). The correct number is 5/22 strains.

2. line 216-217: please indicate the thfT strains in supplementary table 1

As requested, we have modified supplementary table 1 to include the *thfT* strains.

3. line 76: It would be good to introduce dhfG and dhfF here already (now they come a bit out of the blue later on) and possibly also acquired genes (other than thfT) that confer SMX resistance (if known)

As requested, we have modified the text to introduce dfrF and dfrG. We have also clarified that prior to our discovery of *thfT* there are no known acquired SMX resistance genes in GAS. Lines 84-88: "In GAS, individual resistance to TMP and SMX has been attributed to mutation of the target enzymes (Dyr and FolP, respectively), or the acquisition of TMP-resistant variants of Dyr (e.g. DfrF and DfrG^{6,12-14}). However, no horizontally-acquired SMX genes have been described in GAS and the combinations of SMX and TMP resistance that lead to SXT resistance remain unclear. "